# Interplay between BRCA1 and GADD45A and Its Potential for Nucleotide Excision Repair in Breast Cancer Pathogenesis

**DOI:** 10.3390/ijms21030870

**Published:** 2020-01-29

**Authors:** Sylwia Pietrasik, Gabriela Zajac, Jan Morawiec, Miroslaw Soszynski, Michal Fila, Janusz Blasiak

**Affiliations:** 1Department of Molecular Genetics, Faculty of Biology and Environmental Protection, University of Lodz, 90-236 Lodz, Poland; sylwia.pietrasik09@gmail.com (S.P.); gzajac@vp.pl (G.Z.); 2Department of Surgical Oncology, Nicolaus Copernicus Memorial Regional Cancer Center, 93-513 Lodz, Poland; jmorawiec@op.pl; 3Department of Molecular Biophysics, Faculty of Biology and Environmental Protection, University of Lodz, 90-236 Lodz, Poland; miroslaw.soszynski@biol.uni.lodz.pl; 4Department of Neurology, Research Institute, Polish Mother Memorial Hospital, 93-338 Lodz, Poland; michalfila@poczta.onet.pl

**Keywords:** breast cancer, nucleotide excision repair, NER, BRCA1, GADD45A, DNA demethylation

## Abstract

A fraction of breast cancer cases are associated with mutations in the *BRCA1* (BRCA1 DNA repair associated, breast cancer type 1 susceptibility protein) gene, whose mutated product may disrupt the repair of DNA double-strand breaks as BRCA1 is directly involved in the homologous recombination repair of such DNA damage. However, BRCA1 can stimulate nucleotide excision repair (NER), the most versatile system of DNA repair processing a broad spectrum of substrates and playing an important role in the maintenance of genome stability. NER removes carcinogenic adducts of diol-epoxy derivatives of benzo[α]pyrene that may play a role in breast cancer pathogenesis as their accumulation is observed in breast cancer patients. NER deficiency was postulated to be intrinsic in stage I of sporadic breast cancer. BRCA1 also interacts with GADD45A (growth arrest and DNA damage-inducible protein GADD45 alpha) that may target NER machinery to actively demethylate genome sites in order to change the expression of genes that may be important in breast cancer. Therefore, the interaction between BRCA1 and GADD45 may play a role in breast cancer pathogenesis through the stimulation of NER, increasing the genomic stability, removing carcinogenic adducts, and the local active demethylation of genes important for cancer transformation.

## 1. Introduction

Breast cancer, as is the case for many other, if not all, cancers, is underlined by genomic instability, which results from excessive DNA damage and/or an impaired DNA damage response (DDR). DDR is a multi-pathway, complex reaction of the cell to DNA damage, in which DNA repair plays a pivotal role.

About 5% of breast cancer cases are associated with pathogenic variants of the *BRCA1* (BRCA1 DNA repair associated, breast cancer type 1 susceptibility) and *BRCA2* genes (Figure 1) [1]. The presence of such variants increases the lifetime risk of breast cancer by 40–90% [2]. The protein products of both genes are involved in genome protection [3]. Several genome-protective functions have been attributed to BRCA1, including transcription regulation, DNA repair, chromatin remodeling, and ubiquitin ligation [4]. BRCA1 functions as a tumor suppressor due to its role in the maintenance of genomic stability via its multiple roles in the cellular response to DNA double-strand breaks (DSBs, see next sections). That role includes its involvement in cell cycle control, chromatin remodeling, homologues recombination repair (HRR), and non-homologues end-joining (NHEJ) [4]. Although not directly proven, it is accepted that the inefficient repair or misrepair of DSBs by HRR or NHEJ may be causal for breast cancer, at least for cases that are associated with BRCA mutations (reviewed in [5]). Emerging evidence suggests that not only HRR, firstly reported to link breast cancer with BRCA mutations, but also NHEJ and especially its error-prone alternative versions, may play an important role in breast cancer pathogenesis [6]. However, the potential role of BRCA1/2 in sporadic breast cancer is not completely clear and it is hypothesized that haploinsufficiency of these two genes may be enough to initiate breast carcinogenesis or that these two genes are not involved in sporadic breast cancer [6]. Therefore, further studies are needed to link the role of BRCA1 in maintaining genomic stability with breast cancer.

Breast cancer can also be a part of hereditary cancer-related syndromes, including Li-Fraumeni syndrome, Cowden syndrome, and Peutz-Jeghers syndrome, as well as hereditary diffuse gastric cancer [7,8,9,10]. Therefore, variants of genes other than *BRCA1/2* may increase the breast cancer risk (Figure 1). These include *TP53* (tumor protein p53), *PTEN* (phosphatase and tensin homologue), *STK11* (serine/threonine kinase 11), *CDH1* (cadherin 1), *CHEK2* (checkpoint kinase 2), *PALB2* (partner and localizer of BRCA2), *NBN* (Nibrin), *ATM* (ataxia telangiectasia mutated), *BRIP1* (BRCA1 interacting protein C-terminal helicase 1), and *BARD1* (BRCA1 associated RING domain 1) [11,12].

Not all familial breast cancer cases can be explained by the changes in genetic factors identified to date and changes in the heritable epigenetic profile also play a role.

## 2. BRCA1—A Protein of DNA Damage Response and A Tumor Suppressor

BRCA1 is a nuclear phosphoprotein of 1863 aa and tumor-suppressor, encoded by the *BRCA1* gene located in 17q21.

Mutations in the *BRCA1* gene may result in unregulated cell growth and tumor development (reviewed in [13]). BRCA1 contains three major domains: the RING (really interesting new gene) domain at the N-terminus, with ubiquitin-conjugating activity; the BRCT (BRCA1 C-terminal) domain at the C-terminus that can act as a transcriptional activation domain; and a central part with a large unstructured region encoded by exons 11–13 [14] (Figure 2). BRCT and RING are implicated in the interaction between BRCA1 and other proteins and their mutations are found in breast cancer [15,16].

Early studies on BRCA1 functions showed that BRCA1-deficient cells exhibited genomic instability, including chromosomal aberrations [17,18,19,20]. Those reports suggested that BRCA1 may be involved in DDR induced by DSBs. In fact, BRCA1 is phosphorylated in response to DSBs and is recruited to the DSB sites by the phosphorylated Abraxas 1 protein, where it can form several, mutually exclusive, complexes, functioning at different times in DDR and in different cell cycle phases [21]. The involvement of BRCA1 in DDR is not completely known, but it associates with RAD51 (RAD51 recombinase), the main human recombinase, implying a pivotal role in DSB repair by homologous recombination (HRR) [22]. This suggests a role of BRCA1 in DSB repair by HRR. However, BRCA1 can be involved in the other main DSB repair system, non-homologous end joining (NHEJ), but this involvement is far less known than the role of BRCA1 in HRR. BRCA1 also functions in DNA damage-responsive checkpoint control and the resolving of DNA replication inhibited by DNA damage [23]. Another DDR-related function of BRCA1 is its involvement in chromatin remodeling through its association with the SWI/SNF (Switch/sucrose non-fermentable) complex [24].

Other than HRR, DNA repair activities have been reported for BRCA1. Saha et al. showed that BRCA1 transcriptionally upregulated several genes of base excision repair (BER), including *OGG1* (8-oxoguanine DNA glycosylase), *NTHL1* (Nth like DNA glycosylase 1), and *APEX1* (apurinic/apyrimidinic endodeoxyribonuclease 1) [25,26]. In that effect, BRCA1 stimulated the activity of the transcription factor Oct-1 (octamer-binding transcription factor 1).

BRCA1 is also involved in the repair of interstrand crosslinks (ICLs), which represent one of the most, if not the most, serious DNA damage. The Fanconi anemia network is a group of proteins, which, when mutated, result in Fanconi anemia (FA), a genetic disorder affecting many parts of the body and associated with an increased frequency of cancer [27]. FA proteins are involved in ICL repair, but accurate ICL repair also requires HRR and nucleotide excision repair (NER), and other pathways may play a role. An HRR-independent role of BRCA1 in ICL repair can also be considered, presumably by NHEJ [28].

## 3. Nucleotide Excision Repair

Nucleotide excision repair (NER) is perhaps the most versatile DNA repair pathway recognizing and processing damage caused by concurrent disturbances in the DNA secondary structure determined by distorted base pairing and the chemical modification of nucleotides (reviewed in [29]). Nucleotide excision repair has been categorized into global genome NER (GG-NER) and transcription-coupled NER (TC-NER) (Figure 3).

GG-NER checks the entire genome for DNA distortion, whereas TC-NER is activated by RNA polymerase II stalling at the site of distortion, but damages in the non-transcribed strand of active genes are processed by GG-NER. In GG-NER, XPC (XPC complex subunit, DNA damage recognition and repair factor, Xeroderma pigmentosum group C) forms a complex with CETN2 (centrin 2) and hRAD23B (RAD23 homolog B, nucleotide excision repair protein) (Figure 3). It recognizes and preferentially binds the site of DNA damage where base pairing is disrupted. XPC-CENT2-hRAD23 can be assisted by the DDB (damage specific DNA binding protein) complex consisting of DDB1 and DDB2. DDB is present in some lesions, particularly UV-induced cyclobutane pyrimidine dimers, but it is likely not essential for NER and helps to recognize NER substrates within the chromatin [30]. In TC-NER, DNA damage stalls RNA polymerase II that interacts with CSB (ERCC excision repair 6, chromatin remodeling factor) and the CSA (ERCC excision repair 8, CSA ubiquitin ligase complex subunit)-CSB complex. These interactions likely result in backward translocation of the complex (backtracking) and make space for other NER factors. The transcription initiation factor of RNA polymerase II H (TFIIH) is recruited after damage recognition in both GG- and TC-NER. XPG (ERCC excision repair 5, endonuclease), a single-strand-specific DNA endonuclease, is recruited to the complex, either separately or as part of TFIIH that releases CAK (cyclin dependent kinase 7). TFIIH further unwinds DNA in the vicinity of the damage that is then verified by the concerted action of XPD (ERCC excision repair 2, TFIIH core complex helicase subunit), XPB (ERCC excision repair 3, TFIIH core complex helicase subunit), and XPA (DNA damage recognition and repair factor). XPA strongly binds to chemically altered nucleotides in a single DNA strand. The other, undamaged strand is coated by replication protein A (RPA). XPA recruits the XPF (ERCC excision repair 4, endonuclease catalytic subunit)-ERCC1 (ERCC excision repair 1, endonuclease non-catalytic subunit) that makes a cut on the damaged strand 5′ to extract the damage. Then, a 3′ cut is made by XPG that results in the excision of a single-stranded fragment of DNA containing the damage. A PCNA (proliferating cell nuclear antigen) ring-shaped trimer is loaded onto DNA recruiting DNA polymerase δ or ε to synthesize new DNA to fulfill the gap, and DNA ligase 1 or 3 catalyzes the formation of a phosphodiester bond to seal the DNA.

The ERCC1 protein has four isoforms that have different affinities to XPF, resulting in its different activity and, as a consequence, different efficacy of NER [31]. This is important for the reaction of cancer cells to chemotherapy with drug-inducing DNA damages that are substrates for NER, either singly or in combination with other DNA repair pathways (see below). There are some conflicting results on the value of ERCC1 as a prognostic marker in the therapy of cancer, especially non-small cell lung cancer (NSCLC) treated with *cis-*diamminedichloroplatinum (II) (cisplatin), in which resistance to this drug is a common problem [32,33,34]. These conflicting results may originate from the use of antibodies that were not suitable for distinguishing different isoforms of ERCC1, as shown by Friboulet et al. [35]. These authors found that only one—the ERCC1 isoform 202—out of four ERCC1 isoforms had a full functional capacity in NER and could be associated with cisplatin resistance in NSCLC [31]. This was confirmed by Wang et al., who showed that ERCC1 could be a prognostic marker in advanced NSCLC treated with platinum-based drugs [36].

NER may remove several unrelated DNA lesions in a single act if they all are located on the same strand and the other strand is undamaged, to serve as a template for DNA synthesis. To do so, in GG-NER, the repair machinery must have access to damaged DNA in chromatin. This access is granted by the interaction of NER proteins, first of all, the DDB complex, with chromatin components. This interaction results in several modifications of chromatin, including the ubiquitination of core histones and PARP1 (poly(ADP­ribosyl) polymerase 1)­dependent PARylation of chromatin, playing an important role in general DDR [37,38,39]. Mechanisms of chromatin modifications during TC-NER are different from those for GG-NER as transcription itself is associated with changes in the chromatin structure [40].

As mentioned, NER may play an essential role in the removal of interstrand crosslinks (ICLs) [41]. Apart from NER, structure-specific endonucleases, translesion DNA synthesis (TLS), homologous recombination repair (HRR), and Fanconi anemia (FA) proteins have been recognized to be involved in ICL repair.

Defects in NER can be attributed to several hereditary human diseases, including xeroderma pigmentosum (XP). XP, an autosomal recessive genetic disorder, is characterized by increased sensitivity to UV, resulting in skin cancers underlined by a decreased efficacy to repair UV-induced photoproducts [42]. The great majority of XP cases are associated with mutations in NER genes involved in the recognition and excision of UV-related photoproducts.

## 4. Nucleotide Excision Repair in Breast Cancer

Decreased DNA repair synthesis was observed in peripheral blood lymphocytes of breast cancer patients upon irradiation with UV-C [43]. Interestingly, such reduced DNA repair synthesis was also observed in healthy women who had a first-degree relative with breast cancer [44]. However, these studies did not include a life-time follow-up, so it could not be excluded that the subjects of interest would develop breast cancer later. Moreover, mechanisms responsible for NER deficiency in breast cancer may start to operate before clinical manifestation of the disease and not necessarily be associated with breast cancer. No BRCA1 status was investigated in these studies.

Latimer et al. observed that 19 out of 19 stage I sporadic breast cancer samples exhibited NER deficiency, compared with 23 samples from normal epithelial tissues obtained from healthy individuals [45]. The nucleotide excision repair capacity in that work was assayed on the basis of DNA repair synthesis. A microarray analysis performed in that study revealed that 20 canonical NER genes were downregulated on the mRNA level and the reduced expression of 19 of them was confirmed by a more sensitive RNase protection assay. The level of protein products of five of those genes was decreased. Although the authors did not perform any NER-specific functional assay, so the true NER activity was not assayed, the diminished repair DNA synthesis and downregulation of some NER genes, both on mRNA and protein levels, suggest that DNA-damaging agents producing NER substrates, including those used as anticancer drugs, can produce more DNA damage in the initial stage of breast cancer cells than in normal breast epithelium. This is important for chemotherapeutic strategies in breast cancer as many of them include DNA-damaging agents producing not only substrates for NER, but also other DNA damages, repaired with a complex pathway with the involvement of NER. Cisplatin and other platinum-based compounds induce mainly intra- and interstrand crosslinks with DNA. The former are repaired by canonical NER and the latter are removed by complex DNA repair pathways, including homologous recombination, the Fanconi anemia pathway, and NER (reviewed in [46]). However, the anticancer action of cisplatin is not limited to its DNA-damaging effects.

Association between breast cancer occurrence and genotypes of various polymorphisms of NER genes has been reported in many case-control studies and several meta analyses [47,48,49,50,51]. Polymorphisms in the *XPD* and *XPG* genes have been associated with higher concentrations of cyclobutane pyrimidine dimers and polycyclic aromatic hydrocarbons in breast cancer samples compared with non-cancer controls [52,53,54,55,56].

Benzo[α]pyrene (BaP) is a polycyclic aromatic hydrocarbon whose diol-epoxide derivatives (BPDEs) form carcinogenic adducts, with DNA binding preferentially to the exocyclic 2-amino position of guanine [57]. BPDEs can also form adducts with proteins, but their role in cancer transformation is not completely clear [58]. BPDE adducts are frequently investigated in tobacco smoking-related carcinogenesis that also includes breast cancer [59].

BPDE induced more chromatid breaks in lymphocytes from breast cancer patients than controls, suggesting that the patients were more sensitive to BPDE-related carcinogenic events [60]. These changes were more pronounced in young (<45 years) women and ever smoker and alcohol drinkers. In addition, this effect was modulated by the polymorphism of the *GTT1* (glutathione S-transferase theta-1) gene, whose product plays a role in antioxidant defense and cancer transformation.

A greater susceptibility to BPDE adducts in breast cancer patients may be underlined by their impaired repair. To test this hypothesis, Motykiewicz et al. analyzed the level of BPDE-DNA adducts in lymphoblasts from sisters discordant for breast cancer [61]. They did not observe any difference in the level of BPDE-DNA adducts between patients and controls immediately after BPDE treatment, but the level of adducts was higher in affected sisters after a 4-h incubation period in BPDE-free medium to allow for DNA repair. Cells from non-affected sisters removed BPDE-DNA adducts almost two times more effectively than cells from affected sisters. In a very similar study, Kennedy et al. showed that the DNA repair capacity in lymphoblastoid cells incubated with BPDE in breast cancer patients was lower than in non-cancer subjects [62].

Using a mouse model, Guo et al. showed that BaP increased the migration and invasion of breast cancer cells in vitro and in vivo [63]. These effects were underlined by upregulated ROS-induced MAPK1 mitogen-activated protein kinase 1 (ERK) signaling and the activation of matrix metalloproteinases 9.

## 5. BRCA1 in Nucleotide Excision Repair in Breast Cancer

The p53 tumor suppressor protein plays a multifunctional regulatory role in several physiological processes, including the cell cycle, DNA repair, and apoptosis (reviewed in [64]). Its gene, *TP53*, belongs to the most frequently mutated genes in cancer [65]. p53 functions as a tumor suppressor, mainly due to its role as a transcription factor of several hundreds of genes essential for cancer transformation, but its direct interaction with other proteins may also be important for that role [66].

Ford and Hanawalt investigated mutations in the *TP53* gene and their association with the extent of UV-induced DNA damage and repair in primary human skin fibroblasts obtained from patients with Li-Fraumeni syndrome that frequently contain early onset breast cancer [67]. The cells with mutations in both alleles of the *TP53* gene were less susceptible to UV-induced toxicity and UV-induced apoptosis than their heterozygous counterpart. These cells removed cyclobutane pyrimidine dimers from the genome less efficiently than cells with heterozygous mutations or normal cells. However, the cells with mutated alleles preferentially repaired the transcribed strand of the repaired genes. These results show that p53 is involved in the regulation of both GG-NER and TC-NER. However, these studies did not show the exact mechanism of this involvement and several pathways can be considered [68]. Later works, mainly from Ford’s lab, showed that p53 regulated damage recognition in NER targeting DDB2 and XPC (reviewed in [68]).

Fan et al. used the breast cancer cell line MCF-7 with the normal p53 protein and its two sublines with p53 disruption [69]. They observed that p53-deficient sublines were more sensitive to cisplatin, but not to other DNA-damaging agents that did not produce substrates for NER. These authors suggested that these cells had defects in G2/M checkpoint control, NER, or both. These results confirm that p53 can regulate NER in breast cancer.

Hartman and Ford showed that BRCA1 specifically enhanced GG-NER and induced the p53-independent expression of *XPC*, *DDB2*, and *GADD45A* (growth arrest and DNA damage-inducible protein GADD45 alpha) [70]. These authors found that BRCA1 selectively affected GG-NER and the repair of non-transcribed strands in cells exposed to UV in a p53-independent manner. A striking feature of these studies was the difference in the influence of BRCA1 overexpression on the efficacy of repair of the two main DNA damages induced by UV—cyclobutane pyrimidine dimers (CPDs) and 6-4 photoproducts ((6-4)PPs). BRCA1 overexpression increased the repair of CPDs by about 30%, whereas the repair of (6-4)PPs was not affected, but it should be noted that this kind of damage was completely removed independently of the p53 and BRCA1 status. It is known that the repair of CPDs located on non-transcribed strands and in GG-NER can be more than half as efficient in p53-deficient cells than in their p53 wildtype counterparts [67,71,72,73]. This may suggest that such DNA damages are processed by NER, with no critical involvement of BRCA1 and p53. The initial phase of NER relies on two steps: the recognition of secondary structural disturbance in DNA and verification of its chemical alteration. These steps are largely executed by the hRAD23/XPC complex assisted by other proteins that may depend on the type of DNA damage. It is possible that UV-induced photoproducts do not need BRCA1 and p53 in these initial steps of NER. The involvement of UV exposure in breast cancer pathogenesis is not completely clear, but it seems that some sunscreens may induce a more pronounced effect than UV itself [74,75]. Moreover, CPD and (6-4)PP have significantly different structures, which are differentially processed in direct reverse repair in non-placental organisms with a more complex repair reaction for (6-4)PP than CPD. In humans, this may be reflected in the different recognition of these two DNA damages, resulting in different processing.

## 6. DNA Methylation in Breast Cancer

DNA methylation plays an important role in cancer pathogenesis, but the relationship between hypomethylation and hypermethylation in cancer is not completely clear and seems to depend on the tumor type and the stage of its development [76]. Frequently, cancer-linked hypomethylation is more common than hypermethylation, resulting in a decreased all-over content of 5-methylcytosine (5mC) in many human tumors [77]. As promoter methylation may be linked with gene silencing, hypermethylation of a tumor suppressor can be associated with an increased risk of tumor formation and cancer progression [78]. The oncogenic role of DNA hypomethylation is less evident. One of its mechanisms may be the activation of “cancer germline” (“cancer-testis”) genes, a group of germline-specific genes that use DNA methylation to suppress their expression in somatic tissues [79]. These genes influence several pathways in processes important for cancer transformation, including cell proliferation, angiogenesis, genome maintenance, metastasis, apoptosis, and tumor metabolism (reviewed in [80]). Studies on rodents fed with a methyl-deficient diet support the role of hypomethylation in cancer transformation (reviewed in [81]). In general, hypomethylation may play a role in cancer through targeting high- and moderate-copy DNA repeats, leading to genome instability or unique DNA sequences, which results in an increase of the expression of cancer-promoting genes [76].

Hypomethylation was observed in satellite DNA in 40%–90% of breast adenocarcinomas [82]. No association between DNA methylation in peripheral blood lymphocytes and breast cancer was reported [83]. However, several correlations between the methylation of specific genes in whole blood and breast cancer occurrence were observed [84]. An increased methylation of *BRCA1* is often found in the peripheral blood of breast cancer patients compared with cancer-free controls.

Many types of human cancers present global genomic hypomethylation, including prostate tumors, hepatocellular carcinoma, B-cell chronic lymphocytic leukemia (B-CLL), cervical cancer, and others [85,86,87,88]. However, these relationships cannot be generalized, as Kushawa et al. showed that both hypomethylation and hypermethylation are common in B-CLL and there is an interplay between them that contributes to cancer development [89]. Additionally, Hakkarainen et al. reported a higher extent of global DNA methylation in breast cancer compared with samples from lobular breast carcinoma and benign breast tumors [90]. A small increase in the global levels of 5mC was found in ductal carcinomas. In contrast to the latter results, significant global hypomethylation in breast ductal carcinomas compared with normal breast tissue or benign breast tumors was observed in two large-scale studies [91,92].

Another problem is the DNA methylation status of primary breast cancer and its corresponding metastases or, in general, the difference between methylation in metastatic and non-metastatic cancers. We have recently shown that 5-aza-2’-deoxycytidine (5-aza-dC), a DNA demethylation agent, induced different effects in metastatic and non-metastatic colon cancer cell lines [93]. Kirn et al. observed methylation of the promoter of the estrogen receptor 1 (ESR1) gene in 19 out of 25 primary breast cancer cases, but this fraction increased to 24/25 in metastatic disease [94].

Many studies aim to seek a correlation between breast cancer occurrence, its clinico-pathological type, stage of progression, kind of therapy, and the DNA methylation status of the whole genome and/or specific gene(s), but the results obtained so far suggest that this correlation may depend on many variables. Therefore, to look for the correlation, possible confounding factors should be identified and included in an adjusted analysis to answer how a change in the DNA methylation profile may contribute to cancer transformation.

## 7. GADD45A and Its Role in Breast Cancer

The growth arrest and DNA damage-inducible 45 alpha (GADD45A) gene plays a role in the regulation of the cellular reaction to stress as its transcript levels increase in response to stressful growth arrest conditions and DNA-damaging agents. The protein encoded by this gene responds to environmental stresses by mediating activation of the p38/JNK (c-Jun N-terminal kinase) pathway via the MTK1 (methylthioribose kinase 1)/MEKK4 (MEK kinase 4) kinase. Alternatively spliced transcript variants encoding distinct isoforms have been found for this gene [95].

The *GADD45A* gene codes for a 165 aa protein, the GADD45A protein, localized in the nucleus and ubiquitous in normal tissues. The level of GADD45A is related to the cell cycle and is highest in the G1 phase, with a substantial reduction in S [96,97]. It is degraded in the proteasome on ubiquitination and the activation of PKCdelta (protein kinase delta) may result in GADD55A deubiquitination, protecting it from degradation [98].

GADD45A plays several important functions in the cell, including cell growth suppression; mediating cell cycle arrest at G2/M; apoptosis induction; and interaction with p53, CDK1 (cyclin dependent kinase 1, Cdc2), and cyclin B1 (Figure 4, reviewed in [99]). These functions underline an important role of GADD45A in maintaining genomic stability, DDR, and cancer transformation. In fact, the protective role of GADD45A in DNA damage-induced tumorigenesis is a main biological function of this protein that has been demonstrated so far [100,101,102]. However, molecular pathways underlying this protective role of GADD45A in cancer are not completely known.

The important role of GADD45A in cancer transformation was also confirmed in breast cancer. A higher expression of *GADD45A* was observed in breast cancer samples compared with adjacent non-cancerous tissues [103]. Tront et al. observed that the loss of GADD45A accelerated mammary tumor formation driven by Ras activation in mice and that these tumors showed an increasing growth rate and were characterized by a more aggressive phenotype [104]. Moreover, they observed a decrease of apoptosis that was associated with the activation of mitogen-activated protein kinase 8 (MAPK8), as well as a decrease in Ras-induced senescence that was correlated with a decreased activation of p38. These authors presented the tumor-suppressive role of GADD45 in Ras-driven breast carcinogenesis through the induction of stress-sensitive JNK and p38 kinases. However, in their next work, Tront et al. showed that GADD45A promoted Myc-driven breast carcinogenesis by a negative regulation of MMP10 via GSK3 β/β-catenin signaling that resulted in enhanced tumor vascularization and growth [105]. These two works suggest that GADD45A may function as either a tumor promotor or suppressor, depending on the kind of oncogenic stress, and these two functions are mediated by different signaling pathways. Some associations between variability of the GADD45A gene and the occurrence of breast cancer with no BRCA1/2 mutation (sporadic and familial) were found in a large cohort study performed in a Chinese population [106]. Similar results were also obtained in nearly 100 French Canadian families without BRCA1/2 mutations [107].

Tront et al. correlated *GADD45A* expression with the hormone receptor status in breast cancer samples and normal tissues [108]. They observed that normal breast tissue was characterized by low levels of GADD45A, but high levels of this protein were observed in luminal A and B types of breast cancer. Triple negative breast cancer was associated with low levels of GADD45 or its absence. Therefore, the expression of *GADD45A* may be a prognostic marker in breast cancer and GADD45A may thus be considered a target in breast cancer therapy.

Fabbro et al. showed that BRCA1 binding partner BARD1 reduced the transcriptional activity of BRCA1 mediated by BRCA1/BARD1 E3 ligase activity that was disrupted by the 61C>G mutation in the *BRCA1* gene [109].

It was shown that 5-azaC induced the expression of *GADD45* in breast cancer cell lines, with little or no expression in cell lines derived from normal breast epithelium [110]. Therefore, GADD45 can be considered a methylation-sensitive gene in breast cancer.

## 8. BRCA1 Stimulates GADD45A-Mediated NER and Active DNA Demethylation

The DNA damage-induced transcription of *GADD45A* is mediated by both p53-dependent and -independent mechanisms (Figure 5) [111]. In the absence of DNA damage, p53-dependent regulation of *GADD45A* is inhibited by BRCA1 that interacts with ZNF350 (zinc finger protein 350) [109,112]. However, when DNA damage is present, the BRCA1/ZNF350 complex may activate *GADD45A* by recruiting p53 and other *GADD45A*-targeting transcription factors [109].

Several works have shown that BRCA1 strongly activates *GADD45* in a p53-independent fashion [113]. The normal transactivation activity of BRCA1 is required to activate *GADD45*, as tumor-derived mutants of *BRCA1* did not do so [114]. However, the exact mechanism of BRCA1-induced activation of *GADD45A* may be complex as at least three sequence motifs in the GADD45A may play a role in this effect. These are as follows: a BRCA1 binding site located in the first intron of the gene, a ZBRK1 (Zinc finger and BRCA1 interacting protein with a KRAB domain 1) binding site in the third intron, and OCT1 and CCAAT-box elements located between −121 and −75 in the promoter region [113,114,115]. Moreover, BRCA1 associates with transcription factors Oct-1 and NFYA (nuclear transcription factor Y subunit alpha) that bind the OCT1 and CAAT motifs [69].

When BRCA1 induces *GADD45A* in breast cancer, it may lead to apoptosis through the JNK pathway and interaction with MTK1/MEKK4 [113]. GADD45A is the only protein of the GADD45 family that is induced by ionizing radiation—a putative factor of breast carcinogenesis—in human cells with normal p53 [116]. However, such induction is blocked by MDM2 (MDM2 proto-oncogene) that forms an autoregulatory loop with p53 [117]. On the other hand, p53 may be involved in the regulation of *GADD45A* through the ATM kinase [118,119]. Therefore, interaction between GADD45A and p53 may be important for the role of GADD45A in the reaction of breast epithelial cells to ionizing radiation and consequently, breast carcinogenesis.

The involvement of GADD45A in DDR may also be underlined by the role this protein plays in DNA repair. Simit et al. were the first to show that GADD45A associated with PCNA when GADD45A was induced after DNA damage [120]. These authors used an in vitro assay to show an approximately threefold reduction of NER in a nuclear extract containing antibodies to GADD45A, but a 3-5-fold increase in NER was observed when a recombinant GADD45A was added to the extract. This study revealed the role of GADD45A in NER, but it did not solve whether that role was direct or indirect. GADD45A might support the formation of the PCNA-containing repair complex or associate with the DNA polymerase δ repair complex. These studies also showed that the inhibition of *GADD45A* with antisense RNA resulted in a reduced survival of two colon carcinoma RKO cell lines exposed to UV and the authors concluded that that effect was linked to an adverse influence of GADD45A reduction on some aspects of repair-mediated survival. However, this work was criticized soon after it appeared. Kazantzev and Sancar, as well as Kearsey et al., provided arguments that GADD45A did not modulate human excision nuclease or have an effect on the repair synthesis in human NER [121,122].

The critics of the work of Smith et al. [120] did not exclude the involvement of GADD45A in NER. Carrier et al. showed that GADD45A directly interacted with the core histones and destabilized DNA-histone complexes after UV irradiation [123]. This effect mediated the stimulation of relaxing and cleavage activity of DNA topoisomerases, which play a role in the structural organization of chromatin. Therefore, GADD45A can detect alterations in chromatin evoked by DNA-damaging factors and change its structure to facilitate its accessibility to cellular proteins, including DNA repair proteins. These results were confirmed by the subsequent work of Smith et., who showed that GADD45A might be involved in the coupling between chromatin remodeling and DNA repair [124]. FOXO3a (forkhead box O3a) was shown to function at the G2/M checkpoint and stimulate DNA repair [125]. These studies also showed that GADD45A was a direct target for FOXO3a that mediated its effect on DNA repair. The interaction between GADD45 and FOXO3a was confirmed in subsequent studies [126]. Maeda et al. showed a higher basal level of p21 (CDKNA1, cyclin dependent kinase inhibitor 1A, WAF1, CIP1) in GADD45A-deficient mice and keratinocytes from these mice were defective in UV-induced NER [127]. However, cells obtained from animals with double p21/GADD45 knockout showed a normal NER in response to UV. These results suggest that GADD45 stimulates NER by the downregulation of p21. However, the role of p21 in the regulation of NER is not completely known and is, in general, controversial (reviewed in [128]). This protein was reported to be downregulated in UV-irradiated cells resulting from p53 degradation induced by DDB2, which assists hRAD23B/XPC in substrate recognition in NER [129]. It was also shown that p21 upregulation inhibited NER DDB2-deficient mice. The p21 protein has the CDK (cyclin-dependent kinase)- and PCNA-binding domains that are responsible for its effect on the cell cycle and replication [130]. It was shown that these domains did not inhibit DNA repair synthesis associated with NER [131]. Moreover, it was reported that cells with double knockout in the *p21* gene showed NER deficiency [132]. Cazzalini et al. proposed another mechanism for the involvement of p21 in NER—interaction with the p300 acetyltransferase, which disrupted the connection between it and PCNA promoting NER [133]. In addition, p21 may also change the interaction between p300 and XPG [134]. Therefore, currently, it is difficult to assess how p21 may affect NER in breast cancer and further works should address this issue.

Another activity of GADD45A that can be related to DDR is its involvement in active DNA demethylation (reviewed in [135]). Barreto et al. found that expression of the *GADD45A* gene in human and frog cells reactivated expression of the luciferase reporter gene that was inactivated by promoter methylation [136]. This demethylation was active as it occurred in both dividing and non-proliferating cells. This effect was independent of cell proliferation, but depended on UV-induced stress, as such stress involved global DNA demethylation that was mediated by GADD45A. But how this occurred remained an unanswered question. The authors observed that the knockdown of *GADD45A* was associated with downregulation of the *MLH1* (*MutL* homolog 1) tumor suppressor gene, whose product forms a complex with PMS2 (PMS1 homolog 2) that is involved in DNA mismatch repair (MMR) and is regulated by DNA methylation. Observed downregulation of *MLH1* in GADD45A-deficient cells was confirmed to be associated with increased DNA methylation. Active demethylation can be associated with synthesis-dependent DNA repair and base excision repair (BER) seems to be the most involved in this process, but, conceptually, NER and MMR are equally important (reviewed in [137]).

In summary, a multifaceted role of BRCA1 in breast cancer-related NER may be underlined by the following chain of events (Figure 6). BRCA1 stimulates NER that can play a role in the maintenance of genomic stability that may be p53-related and prevent a loss of heterozygosity in BRCA1+/− cells. NER may remove DNA damages that could play a role in breast carcinogenesis, including BDPE adducts. Barreto et al. showed that DNA repair synthesis required GADD45A, which directly interacted with XPG during active DNA demethylation [136]. These authors postulated that GADD45A is targeted to sites of active DNA demethylation and recruits NER components to excise DNA fragments containing 5mCs. Another important message coming from that research was that a reduced activity of GADD45A may result in increased methylation of the tumor suppressor MLH1, which can contribute to genomic instability that is typical for cancer.

However, many outstanding questions can be asked on the role of GADD45A in active DNA demethylation with the involvement of DNA repair and their answers require further research [138].

## 9. Conclusions and Perspectives

A mutated form of BRCA1 is associated with impaired DNA repair, which, in turn, plays a role in breast cancer pathogenesis. However, the deficient repair of DSBs is thought to mainly result from the association of BRCA1 with RAD51, the main human recombinase and a protein of HRR. Consequently, breast carcinogenesis was associated with unrepaired DSBs induced by ionizing radiation and other DSB-inducing factors. However, exposure of the general population to high-energy gamma radiation is not common enough to be responsible for breast cancer cases that can be attributed to deficient DNA repair. On the other hand, NER is strongly involved in the maintenance of genomic stability whose disruption is associated with cancer transformation. Moreover, NER can be involved in the repair of many kinds of DNA damage that may be causal for many cancers, including breast cancer [139]. Furthermore, it was suggested that NER impairment can be an intrinsic feature of stage I sporadic breast cancer [45]. However, the word “sporadic” meant, in that work, no more than “not familial”. Therefore, both somatic and germ-line mutations in NER and breast cancer genes should be analyzed.

It was shown that BRCA1 stimulated NER and so its deficiencies, also including those associated with its recognized pathogenic variants, can contribute to NER impairment that can play a role in breast cancer pathogenesis.

The direct and specific involvement of GADD45A in NER is controversial and no clear mechanism of such involvement has been proposed. However, some works present strong arguments that GADD45A not only stimulates NER, but also targets NER components to specific sites in the genome to actively demethylate them by the excision of 5mCs. Moreover, the stimulation of NER by GADD45A may be mediated by BRCA1 that links GADD45A, NER, and active demethylation with breast cancer.

The breast cancer specificity of BRCA1-induced NER is not clear and requires further research, including that on hormones and their receptors. Hartman and Ford suggested that the loss of heterozygocity for BRCA1 might lead to an initial decline in NER activity and potentially a loss of ESR1, but these studies require continuation to verify the association between NER and hormonal disturbance in breast cancer [140].

In summary, NER may play a multifaceted role in breast carcinogenesis, but further research is required to clarify this. At present, its role may be summarized as (I) the maintenance of genomic stability, (II) removal of breast cancer-causal DNA damages, and (III) BRCA1- and GADD45-mediated demethylation of breast cancer-related genes.

## Figures and Tables

**Figure 1 ijms-21-00870-f001:**
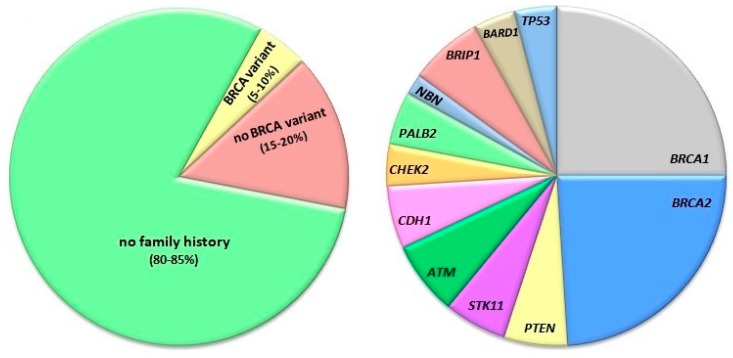
Familial and non-familial breast cancer. The diagram on the left shows the approximate fraction of breast cancer cases with no family history (green) and family history associated with (yellow) or without (brown) the occurrence of BRCA1 (DNA repair associated, breast cancer type 1 susceptibility) and BRCA12 pathogenic variants. The right diagram presents the distribution of pathogenic mutations found in breast cancer cases with family history. Abbreviations are defined in the main text.

**Figure 2 ijms-21-00870-f002:**
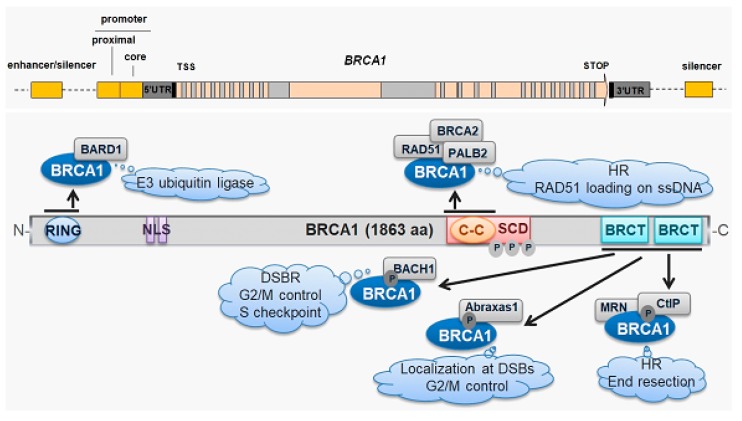
BRCA1 (BRCA1 DNA repair associated) gene and protein. The *BRCA1* gene (upper panel) is located in 17q21.31, contains 24 exons, and encodes the BRCA1 protein (lower panel), which is important for genomic stability (clouds). The RING (really interesting new gene), NLS (nuclear localization signal), coiled-coil (C-C), SCD (serine cluster domain), and BRCT (BRCA1 C-terminal) domains are presented in a linear representation of BRCA1. BRCA1 is phosphorylated in response to DNA double-strand breaks (DSBs) and recruited at the DSB sites by the phosphorylated Abraxas protein. BRCA1 interacts with BARD1 (BRCA1-associated RING domain protein 1) through its N-terminal RING domain to activate its E3 ligase activity. A coiled-coil domain interacts with RAD51 (RAD51 recombinase), the main human recombinase, and its counterpart in PALB2 (partner and localizer of BRCA2). SCD is phosphorylated by the ATM (ataxia telangiectasia mutated)/ATR (ATR serine/threonine kinase) kinases. The BRCT repeats are responsible for multiple interactions of BRCA1 with other proteins. BACH1—BTB domain and CNC homolog 1, CtiP—RB bindfing protein, endonuclease, MRN—MRN complex interacting protein, TSS—transcription start site, UTR—untranslated region, P—a phosphate residue, HR—homologous recombination, and DSBR—double-strand break repair. Dotted lines represent DNA fragments non-essential for the *BRCA1* gene.

**Figure 3 ijms-21-00870-f003:**
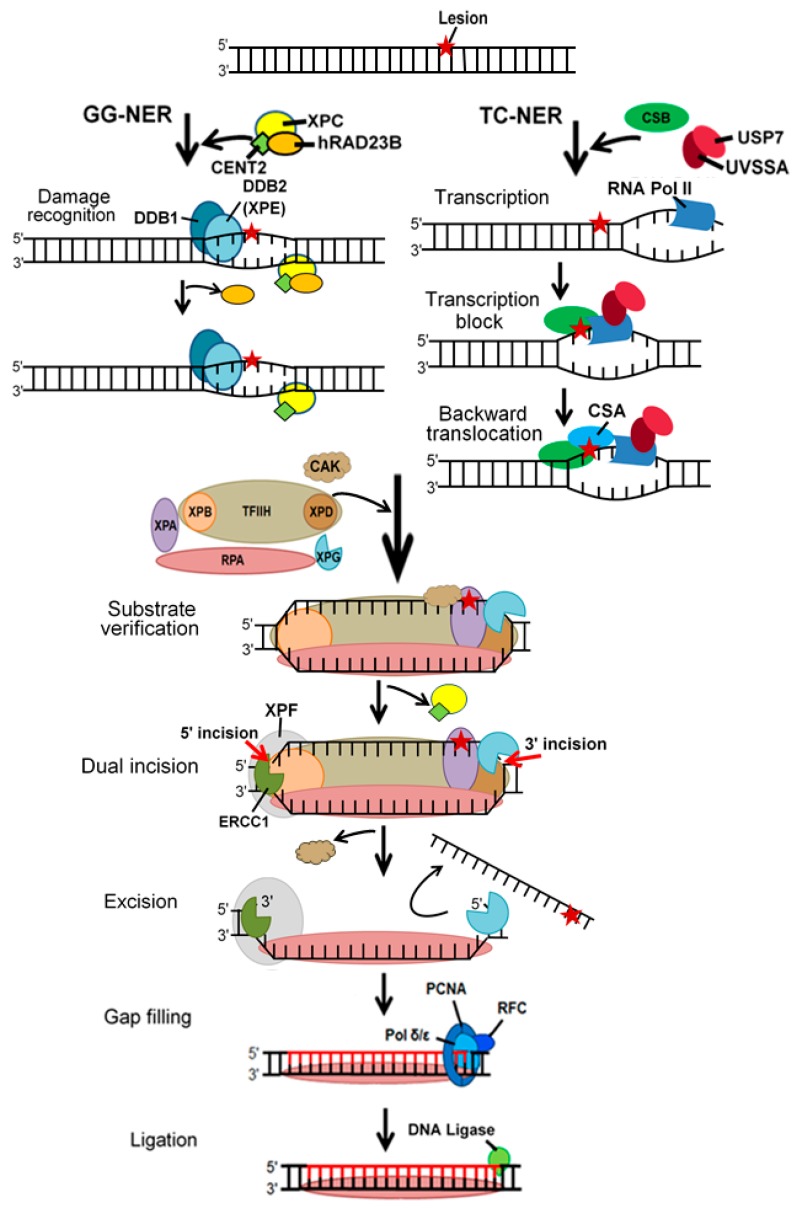
Nucleotide excision repair (NER) in humans. In transcriptionally inactive DNA or non-transcribed strands of active genes, distortion of the DNA helix (red star) is recognized by XPC complexed with hRAD23B and CENT2 in GG-NER. This complex may be supplemented by the DDE complex. hRAD23 leaves the complex upon XPC binding. In the transcription-coupled NER, DNA damage is signalized by the stalling RNA polymerase II (RNA Pol II) during transcription elongation. A complex interaction of UVSSA, USP7, and CSB with RNA Pol II occurs during the elongation, but the stalling of the polymerase induces CSB binding by CSA, likely resulting in backward translocation of the polymerase and making room for the NER machinery. TFIIH, a transcription initiation complex is recruited to the damage in both NER subpathways. XPG, a structure-specific endonuclease, binds to the NER complex. The multiprotein complex TFIIH displays helicase activity and opens DNA around the damage. It also possesses the ATPase activity to DNA 5′→3′ unwind to verify the presence of chemically modified nucleotides in the site of damage by XPD assisted by XPA. Another structure-specific endonuclease, XPF-ERCC1, is directed to the damage by RPA and makes a strand break on its 5′ side, and XPG then makes a cut on the opposite side to the damage. PCNA is loaded onto XPF-ERCC1 and recruits DNA Pol δ/ε to fulfill the gap and DNA ligase 1 or 3 seals the nick, restoring the lacking phosphodiester bond and DNA integrity. Abbreviations are defined in the main text. Black straight arrows represent changes in DNA structure induced by a protein or protein complex (black curved arrows).

**Figure 4 ijms-21-00870-f004:**
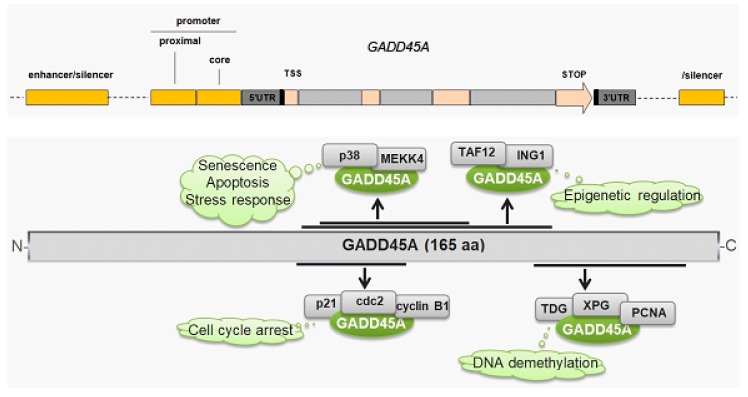
GADD45A (growth arrest and DNA damage-inducible 45 alpha) gene and protein. The *GADD45A* gene is located in 1p31.3 and contains four exons. The GADD45 protein interacts with many proteins to perform many functions in stress signaling, DNA repair, cell survival, cell cycle arrest, senescence, or apoptosis (clouds). TSS—transcription start site, UTR—untranslated region, MEKK4—mitogen-activated protein kinase 4, TAF12—TATA-box binding protein associated factor 12, ING1—inhibitor of growth family member 1, cdc2—cyclin dependent kinase 2, TDG—thymine DNA glycosylase, XPG—xeroderma pigmentosum group, G—complementary protein, and PCNA—proliferating cell nuclear antigen. Dotted lines represent DNA fragments non-essential for the *GADD45A* gene.

**Figure 5 ijms-21-00870-f005:**
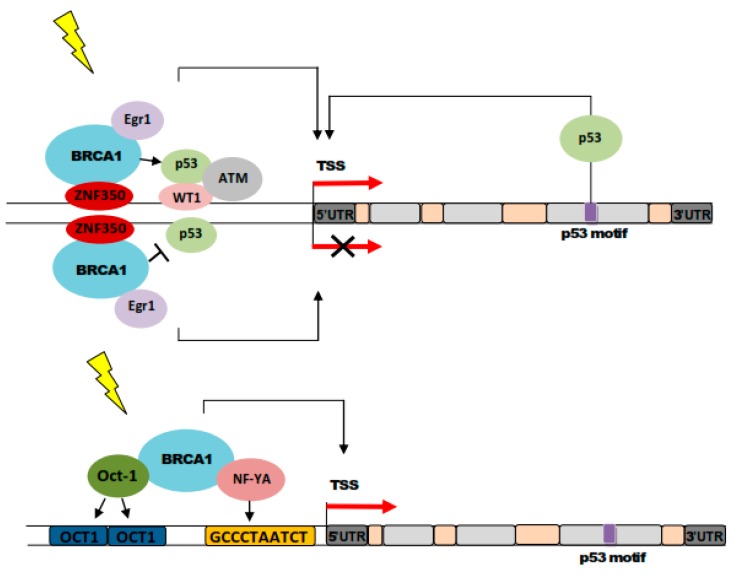
p53-dependent (upper panel) and independent (lower panel) regulation of GADD45A (growth arrest and DNA damage inducible 45 alpha) by BRCA1 (BRCA1 DNA repair associated, breast cancer type 1 susceptibility) in the presence and absence of DNA damage (yellow thunder). BRCA1 stimulates GADD45A when DNA damage is present. The BRCA1/ZNF350 (zinc finger protein 350)/Egr1 (early growth response protein 1) complex activates *GADD45A* by recruiting p53 to the p53 motif located at the *GADD45A* third intron. Ionizing radiation and other DSB-inducing agents promote GADD45A through the ATM (ataxia telangiectasia mutated) targeting p53, which accumulates and stimulates *GADD45A* transcription by binding the WT1 (Wilms tumor gene protein 1) protein, which directly binds the *GADD45A* promoter. In the absence of DNA damage, the BRCA1/ZNF350/Erg1 complex inhibits *GADD45A*. The regulation of *GADD45A* by non-DSB-inducing factors is independent of p53. In such a case, *GADD45A* regulation by BRCA1 is mediated by the interaction of Oct-1 (octamer-binding transcription factor 1) and NF-YA (nuclear transcription factor Y subunit alpha) with two elements present in the GADD45A promoter: OCT1 and CAAT box.

**Figure 6 ijms-21-00870-f006:**
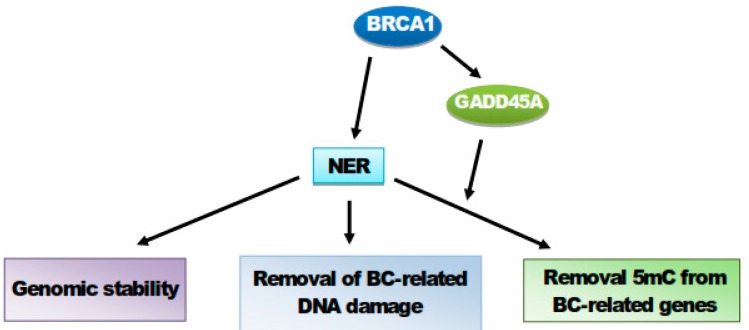
BRCA1 (BRCA1 DNA repair associated, breast cancer (BC) type 1 susceptibility protein) may stimulate nucleotide excision repair (NER) that may prevent or slow down breast carcinogenesis, increasing the genomic stability and removing BC-related DNA damages. Furthermore, BRCA1 may stimulate GADD45A, which can actively demethylate genes important in BC prevention and suppression.

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
