# Peer review of "Interplay between BRCA1 and GADD45A and Its Potential for Nucleotide Excision Repair in Breast Cancer Pathogenesis"

_ijms, 2020, doi:10.3390/ijms21030870_

Round 1
Reviewer 1 Report
The review is interesting and focussed on the role of NER in breast cancer and on the links with BRCA and GADD45
I would suggest an English editing as well as a deeper discussion of some points as detailed here:
1. The fact that in stage I BC a NER deficiency was observed compared to healthy subject would suggest that DNA damagng agents whose lesions are recognized by NER should be active in BC. IS this the case , for example for cisplatin? A comment on this would be helpful.
2. Similarly it is not clear whtehr deficiency means lack or decrease activity. It is also not clear whether "activity" means level of proteins participating to NER or a true activity.
3. NER is complex as clearly outlined by the authors and there are several papers suggesting that the complex ERCC1/XPF, one of the limiting step in NER, has different activity depending on the ERCC1 isoforms associated to XPF. This should be discussed
4. GADD45 is a p53 dependent gene and its levels are increased following damage in a p53 dependent way. This should be mentioned and discussed in the chapter in which BRCA1 GADD45 connection is highlighted.
5. Figure 1. This figure is not very cler. It seems from the left panel that 50% of breast cancer cases are associate dwith BRCA1/2 mutations. Is this correct? The legend should be more clear
Author Response
The review is interesting and focussed on the role of NER in breast cancer and on the links with BRCA and GADD45
Comment I would suggest an English editing…
Answer We have done our best to correct all mistakes, errors and typos we found in the manuscript and improve its style.
Comment …as well as a deeper discussion of some points as detailed here:
The fact that in stage I BC a NER deficiency was observed compared to healthy subject would suggest that DNA damagng agents whose lesions are recognized by NER should be active in BC. IS this the case , for example for cisplatin? A comment on this would be helpful. Similarly it is not clear whtehr deficiency means lack or decrease activity. It is also not clear whether "activity" means level of proteins participating to NER or a true activity.Answer We have added the following fragment to the 4. Nucleotide Excision Repair in Breast Cancer section as a continuation of the sentence “Latimer et al…(lines 187-189 in the original manuscript):
“Nucleotide excision repair capacity in that work was assayed on the basis of DNA repair synthesis. A microarray analysis performed in that study revealed that 20 canonical NER genes were downregulated on the mRNA level and reduced expression of 19 of them was confirmed by a more sensitive RNase protection assay. The level of protein products of 5 of those genes was decreased. Although the authors did not perform any NER-specific functional assay, so the true NER activity was not assayed, diminished repair DNA synthesis and downregulation of some NER genes, both on mRNA and protein levels, suggest that DNA-damaging agents producing NER substrates, including those used as anticancer drugs, can produce more DNA damage in initial stage of breast cancer cells than in normal breast epithelium. This is important for chemotherapeutic strategies in breast cancer as many of them include DNA-damaging agents producing not only substrates for NER, but also other DNA damages, repaired with a complex pathway with the involvement of NER. Cisplatin and other platinum-based compounds induce mainly intra- and interstrand crosslinks with DNA. The former are repaired by canonical NER and the latter are removed by a complex DNA repair pathways, including homologous recombination, Fanconi anemia pathway and NER (reviewed in [46]). However, the anticancer action of cisplatin is not limited to its DNA-damaging effects.”
with new reference:
Dasari, S.; Tchounwou, P.B. Cisplatin in cancer therapy: molecular mechanisms of action. European journal of pharmacology 2014, 740, 364-378, doi:10.1016/j.ejphar.2014.07.025.
Comment 3. NER is complex as clearly outlined by the authors and there are several papers suggesting that the complex ERCC1/XPF, one of the limiting step in NER, has different activity depending on the ERCC1 isoforms associated to XPF. This should be discussed
Answer We have added the following fragment to the 3. Nucleotide Excision Repair section (from line 161 in original manuscript):
“The ERCC1 protein has four isoforms that have different affinity to XPF resulting in its different activity and, in consequence, different efficacy of NER [31]. This is important for the reaction of cancer cells to chemotherapy with drug inducing DNA damages that are substrates for NER, either singly or in combination with other DNA repair pathways (see below). There are some conflicting results on the value of ERCC1 as a prognostic marker in the therapy of cancer, especially non-small cell lung cancer (NSCLC) treated with cis-diamminedichloroplatinum (II) (cisplatin), in which resistance to this drug is a common problem [32-34]. These conflicting results may originate from the use of antibodies that were not suitable to distinguish different isoforms of ERCC1, as showed by Friboulet et al. [35]. These authors found that only one, the ERCC1 isoform 202, out of four ERCC1 isoforms had a full functional capacity in NER and could be associated with cisplatin resistance in NSCLC [31]. This was confirmed by Wang et al. who showed that ERCC1 could be a prognostic marker in advanced NSCLC treated with platinum-based drugs [36].”
with new references:
31 Friboulet, L.; Postel-Vinay, S.; Sourisseau, T.; Adam, J.; Stoclin, A.; Ponsonnailles, F.; Dorvault, N.; Commo, F.; Saulnier, P.; Salome-Desmoulez, S., et al. ERCC1 function in nuclear excision and interstrand crosslink repair pathways is mediated exclusively by the ERCC1-202 isoform. Cell cycle (Georgetown, Tex.) 2013, 12, 3298-3306, doi:10.4161/cc.26309.
32 Hubner, R.A.; Riley, R.D.; Billingham, L.J.; Popat, S. Excision repair cross-complementation group 1 (ERCC1) status and lung cancer outcomes: a meta-analysis of published studies and recommendations. PloS one 2011, 6, e25164, doi:10.1371/journal.pone.0025164.
Ota, S.; Ishii, G.; Goto, K.; Kubota, K.; Kim, Y.H.; Kojika, M.; Murata, Y.; Yamazaki, M.; Nishiwaki, Y.; Eguchi, K., et al. Immunohistochemical expression of BCRP and ERCC1 in biopsy specimen predicts survival in advanced non-small-cell lung cancer treated with cisplatin-based chemotherapy. Lung cancer (Amsterdam, Netherlands) 2009, 64, 98-104, doi:10.1016/j.lungcan.2008.07.014. Sodja, E.; Knez, L.; Kern, I.; Ovcaricek, T.; Sadikov, A.; Cufer, T. Impact of ERCC1 expression on treatment outcome in small-cell lung cancer patients treated with platinum-based chemotherapy. European journal of cancer (Oxford, England : 1990) 2012, 48, 3378-3385, doi:10.1016/j.ejca.2012.06.011.35 Friboulet, L.; Olaussen, K.A.; Pignon, J.P.; Shepherd, F.A.; Tsao, M.S.; Graziano, S.; Kratzke, R.; Douillard, J.Y.; Seymour, L.; Pirker, R., et al. ERCC1 isoform expression and DNA repair in non-small-cell lung cancer. The New England journal of medicine 2013, 368, 1101-1110, doi:10.1056/NEJMoa1214271.
Wang, X.; Zhu, X.; Zhang, H.; Fan, X.; Xue, X.; Chen, Y.; Ding, C.; Zhao, J.; Wu, G. ERCC1_202 Is A Prognostic Biomarker in Advanced Stage Non-Small Cell Lung Cancer Patients Treated with Platinum-Based Chemotherapy. Journal of Cancer 2017, 8, 2846-2853, doi:10.7150/jca.19897.
Comment 4. GADD45 is a p53 dependent gene and its levels are increased following damage in a p53 dependent way. This should be mentioned and discussed in the chapter in which BRCA1 GADD45 connection is highlighted.
Answer: The interconnection between GADD45A and p53 is signalized in the manuscript (lines 370—377 and Figure 5 in the original manuscript) and we would not like to write too much about it not to overload this section with details that are not directly related to the main subject, especially that p53-independent pathway seems to be more linked with it. We have added the following fragment to that part of the manuscript:
“GADD45A is the only protein of the GADD45 family that is induced by ionizing radiation, a putative factor of breast carcinogenesis, in human cells with normal p53 [116]. However such induction is blocked by MDM2 (MDM2 proto-oncogene) that forms an autoregulatory loop with p53 [117]. On the other hand, p53 may be involved in the regulation of GADD45A through the ATM kinase [118,119]. Therefore, interaction between GADD45A and p53 may be important for the role of GADD45A in the reaction of breast epithelial cells to ionizing radiation and consequently breast carcinogenesis”
with new references:
Zhan, Q.; Bae, I.; Kastan, M.B.; Fornace, A.J., Jr. The p53-dependent gamma-ray response of GADD45. Cancer research 1994, 54, 2755-2760. Amundson, S.A.; Lee, R.A.; Koch-Paiz, C.A.; Bittner, M.L.; Meltzer, P.; Trent, J.M.; Fornace, A.J., Jr. Differential responses of stress genes to low dose-rate gamma irradiation. Molecular cancer research : MCR 2003, 1, 445-452. Bishop, A.J.; Hollander, M.C.; Kosaras, B.; Sidman, R.L.; Fornace, A.J., Jr.; Schiestl, R.H. Atm-, p53-, and Gadd45a-deficient mice show an increased frequency of homologous recombination at different stages during development. Cancer research 2003, 63, 5335-5343. Chen, Y.; Yang, R.; Guo, P.; Ju, Z. Gadd45a deletion aggravates hematopoietic stem cell dysfunction in ATM-deficient mice. Protein & cell 2014, 5, 80-89, doi:10.1007/s13238-013-0017-9.
Comment 5. Figure 1. This figure is not very cler. It seems from the left panel that 50% of breast cancer cases are associate dwith BRCA1/2 mutations. Is this correct? The legend should be more clear
Answer I presume that the reviewer meant “right panel”. We have changed the sentence in Figure 1 legend:
“The other diagram presents approximate fraction of breast cancer cases that are associated with mutations in the BRCA1/2 and other genes.”
into
“The right diagram presents the distribution of pathogenic mutations found in breast cancer cases with family history.”
Reviewer 2 Report
The roles of BRCA genes (BRCA1 and BRCA2) in the development of breast cancer is well established. In general, BRCA is mainly implicated in DNA double-strand break (DSB) repair by regulating the homologous recombination repair (HRR). But here the authors argue that BRCA1 is also critical for nucleotide excision repair (NER, this is not a DSB repair system) activity through the action of the transcriptional activity of BRCA1 in the expression of NER genes XPC and DDB2 (as reported by Ford group). However, this is not confirmed by other researchers in the field of NER and an issue of being controversial. Thus it is premature to say that BRCA1 has a "critical" role in the NER as the authors claimed in the title. Besides, GADD45’s role in the regulation of NER seems more indirect through the control of cell cycle inhibitor p21.
As the authors mentioned, NER is the most versatile repair system in mammalian cells, therefore it is conceivable that NER would contribute not only a protective role in the onset of certain cancer but also would play a defensive role for the cancer cells in response to genotoxic agents such as a chemotherapeutic, cisplatin. Likewise, NER may contribute to the physiology of breast cancer but the evidence on the roles of BRCA1 and GADD45 in the regulation of NER is not compelling enough. Therefore, the reviewer believes that it is too premature to write a review paper on the subject. Instead, it should be a perspective. Future works will be needed to prove the hypothesis.
Minor points
Line 101: Full name and function of OCT1 should be addressed here, though they appear in Figure 5. Line 162: For GC-NER, you may want to use GG-NER or TC-NER? Line 165: For ubiquitilation, you may want to use either ubiquitylation or ubiquitination. Line 180, 181: For DNA synthesis, you may want to use either unscheduled DNA synthesis or DNA repair synthesis. Line 248: The audience may be curious about the differential effects of BRCA1 in the repair of CPD and 6-4PP. Also correct 4-6PP to 6-4PP. Line 247-251: What is the purpose of the sentences? Do you mean that p53 and BRCA1 are not critical for NER activity? Line 281-286: The reviewer does not understand why the authors mentioned the importance of various methods in this paragraph. Line 295: The conclusion is somewhat naïve. You may want to add some insights on the results overall. Line 414: The audience may want to hear why the downregulation of p21 affects NER activity.Author Response
Comment The roles of BRCA genes (BRCA1 and BRCA2) in the development of breast cancer is well established. In general, BRCA is mainly implicated in DNA double-strand break (DSB) repair by regulating the homologous recombination repair (HRR).
Answer We have replaced the sentence (line 49 in original manuscript):
“Products of both these genes are involved in DNA repair [3].”
with the following fragment:
“The protein products of both genes are involved in the genome protection [3]. Several genome-protective functions have been attributed to BRCA1, including transcription regulation, DNA repair, chromatin remodeling and ubiquitin ligation [4]. BRCA1 functions as a tumor suppressor due to its role in the maintenance of genomic stability via its multiple role in the cellular response to DNA double-strand breaks (DSBs, see next sections). That role includes its involvement in cell cycle control, chromatin remodeling, homologues recombination repair (HRR) and non-homologues end-joining (NHEJ) [4]. Although not directly proven, it is accepted that inefficient repair or misrepair of DSBs by HRR or NHEJ may be causal for breast cancer at least for these cases that are associated with BRCA mutations (reviewed in [5]). Emerging evidence suggests that not only HRR, firstly reported to link breast cancer with BRCA mutations, but also NHEJ and especially its error-prone alternative versions may play an important role in breast cancer pathogenesis [6]. However, the potential role of BRCA1/2 in sporadic breast cancer is not completely clear and it is hypothesized that haploinsufficiency of these two genes may be enough to initiate breast carcinogenesis or these two genes are not involved in sporadic breast cancer [6]. Therefore, further studies are needed to link the role of BRCA1 in maintaining genomic stability with breast cancer.”
with new references:
Huen, M.S.; Sy, S.M.; Chen, J. BRCA1 and its toolbox for the maintenance of genome integrity. Nature reviews. Molecular cell biology 2010, 11, 138-148, doi:10.1038/nrm2831. Scott, R.J. DNA double strand break repair and its association with inherited predispositions to breast cancer. Hereditary cancer in clinical practice 2004, 2, 37-43, doi:10.1186/1897-4287-2-1-37. Saha, J.; Davis, A.J. Unsolved mystery: the role of BRCA1 in DNA end-joining. Journal of radiation research 2016, 57 Suppl 1, i18-i24, doi:10.1093/jrr/rrw032.
Comment But here the authors argue that BRCA1 is also critical for nucleotide excision repair (NER, this is not a DSB repair system) activity through the action of the transcriptional activity of BRCA1 in the expression of NER genes XPC and DDB2 (as reported by Ford group). However, this is not confirmed by other researchers in the field of NER and an issue of being controversial.
Answer It was not our intention to argue that BRCA1 may be critical for general NER – all we wanted to say was that it might play a role in NER in the context of breast cancer. We have made some changes (see below and revised manuscript) to clarify our point of view.
Comment Thus it is premature to say that BRCA1 has a "critical" role in the NER as the authors claimed in the title.
Answer We have changed the title into:
“Interplay between BRCA1 and GADD45A and its potential for nucleotide excision repair in breast cancer pathogenesis”
Comment Besides, GADD45’s role in the regulation of NER seems more indirect through the control of cell cycle inhibitor p21.
Answer The direct involvement of GADD45A in NER is still a matter of debate and we discuss this issue in lines 386-399 in the original manuscript. The interaction between GADD45A and p21 is discussed in lines 411-415 in the original manuscript.
Comment As the authors mentioned, NER is the most versatile repair system in mammalian cells, therefore it is conceivable that NER would contribute not only a protective role in the onset of certain cancer but also would play a defensive role for the cancer cells in response to genotoxic agents such as a chemotherapeutic, cisplatin. Likewise, NER may contribute to the physiology of breast cancer but the evidence on the roles of BRCA1 and GADD45 in the regulation of NER is not compelling enough. Therefore, the reviewer believes that it is too premature to write a review paper on the subject. Instead, it should be a perspective. Future works will be needed to prove the hypothesis.
Answer We have made several modifications to the manuscript that make it more in-line with this general statement and we suggest to classify it as a “Review/perspective”
Minor points
Comment Line 101: Full name and function of OCT1 should be addressed here, though they appear in Figure 5.
Answer We have corrected all uses of Oct-1/OCT1.
Comment Line 162: For GC-NER, you may want to use GG-NER or TC-NER?
Answer Surely, should be GG-NER.
Comment Line 165: For ubiquitilation, you may want to use either ubiquitylation or ubiquitination.
Answer We have corrected that.
Comment Line 180, 181: For DNA synthesis, you may want to use either unscheduled DNA synthesis or DNA repair synthesis.
Answer We have changed that.
Comment Line 248: The audience may be curious about the differential effects of BRCA1 in the repair of CPD and 6-4PP.
Answer Please see below.
Comment Also correct 4-6PP to 6-4PP.
Answer We have done so.
Comment Line 247-251: What is the purpose of the sentences? Do you mean that p53 and BRCA1 are not critical for NER activity?
Answer We have added the following fragment to that part:
“This may suggest that such DNA damages are processed by NER with no critical involvement of BRCA1 and p53. The initial phase of NER relies on two steps: recognition of secondary structural disturbance in DNA and verification of its chemical alteration. These steps are largely executed by hRAD23/XPC complex assisted by other proteins that may depend on the type of DNA damage. Maybe UV-induced photoproducts do not need BRCA1 and p53 in these initial steps of NER? The involvement of UV exposure in breast cancer pathogenesis is not completely clear, but it seems that some sunscreens may induce a more pronounced effect than UV itself [74,75]. Moreover, CPD and (6-4)PP have significantly different structures, which are differentially processed in direct reverse repair in non-placental organisms with a more complex repair reaction for (6-4)PP than CPD. In humans, this may be reflected in different recognition of these two DNA damages, resulting in their different processing.”
with new references:
Alamer, M.; Darbre, P.D. Effects of exposure to six chemical ultraviolet filters commonly used in personal care products on motility of MCF-7 and MDA-MB-231 human breast cancer cells in vitro. Journal of applied toxicology : JAT 2018, 38, 148-159, doi:10.1002/jat.3525. Barr, L.; Alamer, M.; Darbre, P.D. Measurement of concentrations of four chemical ultraviolet filters in human breast tissue at serial locations across the breast. Journal of applied toxicology : JAT 2018, 38, 1112-1120, doi:10.1002/jat.3621.
Comment Line 281-286: The reviewer does not understand why the authors mentioned the importance of various methods in this paragraph.
Answer We have removed that sentence.
Comment Line 295: The conclusion is somewhat naïve. You may want to add some insights on the results overall.
Answer We have changed the paragraph:
“Many studies aims to seek for a correlation between breast cancer occurrence, type and progression and DNA methylation status of the genome and specific genes, but the only solid general conclusion is that they do matter. Therefore, any action resulting in a change in DNA methylation profile may contribute to cancer transformation positively or negatively.”
into:
“Many studies aim to seek for a correlation between breast cancer occurrence, its clinico-pathological type, stage of progression, kind of therapy and DNA methylation status of the whole genome and/or specific gene(s), but the results obtained so far suggest that this correlation may depend on many variables. Therefore, to look for the correlation, possible confounding factors should be identified and included into an adjusted analysis to answer how a change in the DNA methylation profile may contribute to cancer transformation.”
Comment Line 414: The audience may want to hear why the downregulation of p21 affects NER activity.
Answer We have added the following fragment as a continuation of that paragraph:
“However, the role of p21 in the regulation of NER is not completely known and in general controversial (reviewed in [128]). This protein was reported to be downregulated in UV-irradiated cells resulting from p53 degradation induced by DDB2, which assists hRAD23B/XPC in substrate recognition in NER [129]. It was also shown that p21 upregulation inhibited NER DDB2-deficient mice. The p21 protein has the CDK (cyclin dependent kinase)- and PCNA-binding domains that are responsible for its effect on cell cycle and replication [130]. It was shown that these domains did not inhibit DNA repair synthesis associated with NER [131]. Moreover, it was reported that cells with double knockout in the p21 gene showed NER deficiency [132]. Cazzalini et al. proposed another mechanism of the involvement of p21 in NER – the interaction with the p300 acetyltransferase, disruption the connection between it and PCNA promoting NER [133]. In addition, p21 may also change the interaction between p300 and XPG [134]. Therefore, currently it is difficult to assess how p21 may affect NER in breast cancer and further works should address this issue.”
with new references:
Al Bitar, S.; Gali-Muhtasib, H. The Role of the Cyclin Dependent Kinase Inhibitor p21(cip1/waf1) in Targeting Cancer: Molecular Mechanisms and Novel Therapeutics. Cancers 2019, 11, doi:10.3390/cancers11101475. Stoyanova, T.; Yoon, T.; Kopanja, D.; Mokyr, M.B.; Raychaudhuri, P. The xeroderma pigmentosum group E gene product DDB2 activates nucleotide excision repair by regulating the level of p21Waf1/Cip1. Molecular and cellular biology 2008, 28, 177-187, doi:10.1128/mcb.00880-07. Waga, S.; Stillman, B. Cyclin-dependent kinase inhibitor p21 modulates the DNA primer-template recognition complex. Molecular and cellular biology 1998, 18, 4177-4187, doi:10.1128/mcb.18.7.4177. Soria, G.; Speroni, J.; Podhajcer, O.L.; Prives, C.; Gottifredi, V. p21 differentially regulates DNA replication and DNA-repair-associated processes after UV irradiation. Journal of cell science 2008, 121, 3271-3282, doi:10.1242/jcs.027730. Stivala, L.A.; Riva, F.; Cazzalini, O.; Savio, M.; Prosperi, E. p21(waf1/cip1)-null human fibroblasts are deficient in nucleotide excision repair downstream the recruitment of PCNA to DNA repair sites. Oncogene 2001, 20, 563-570, doi:10.1038/sj.onc.1204132. Cazzalini, O.; Perucca, P.; Savio, M.; Necchi, D.; Bianchi, L.; Stivala, L.A.; Ducommun, B.; Scovassi, A.I.; Prosperi, E. Interaction of p21 CDKN1A with PCNA regulates the histone acetyltransferase activity of p300 in nucleotide excision repair. Nucleic Acids Res 2008, 36, 1713-1722, doi:10.1093/nar/gkn014. Tillhon, M.; Cazzalini, O.; Nardo, T.; Necchi, D.; Sommatis, S.; Stivala, L.A.; Scovassi, A.I.; Prosperi, E. p300/CBP acetyl transferases interact with and acetylate the nucleotide excision repair factor XPG. DNA repair 2012, 11, 844-852, doi:10.1016/j.dnarep.2012.08.001.
Round 2
Reviewer 2 Report
Most of my previous concerns were well addressed in this version of manuscript. Typos: L154 (CETN2), L156 (CTN2) --> CENT2, L333 (B-CCL) --> B-CLLAuthor Response
We have corrected these typos. Thank you for drawing our attention.